# CEO's functional experience and firm performance based on text mining

**Xiaohong Huang** [ID]**[1]\***, **Jiangwei Liu[2]**, **Liangyu Min** [ID]**[3]**, **Qianqian Zeng[4]**, **Jun Zhang[5]**, **Xiaorong Zhang[6]**

**1** School of Logistics and Management Engineering, Yunnan University of Finance and Economics, Kunming, China, **2** School of Computer and Information Engineering, Henan University of Economics and Law, Zhengzhou, China, **3** Faculty of Business Information, Shanghai Business School, Shanghai, China, **4** School of Artificial Intelligence, DongGuan City University, Dongguan, China, **5** NSK(China) Research and Development Co.,Ltd, Kunshan, China, **6** Shanghai University of Finance and Economics, School of Information Management and Engineering, Shanghai, China

\* huangxiaohong@163.sufe.edu.cn

**Data Availability Statement:** Direct URL: https://www.gtarsc.com/ accession numbers: Just register and log in with regular email. Instructions: 1). Register and log in through https://www.gtarsc.com/. 2). Search related data of listed companies.

## Abstract

The impact of a chief executive officer's (CEO's) functional experience on firm performance has gained the attention of many scholars. However, the measurement of functional experience is rarely disclosed in the public database. Few studies have been conducted on the comprehensive functional experience of CEOs. This paper used the upper echelons theory and obtained deep-level curricula vitae (CVs) data through the named entity recognition technique. First, we mined 15 consecutive years of CEOs' CVs from 2006 to 2020 from Chinese listed companies. Second, we extracted information throughout their careers and automatically classified their functional hierarchy. Finally, we constructed breadth (functional breadth: functional experience richness) and depth (functional depth: average tenure and the hierarchy of function) for empirical analysis. We found that a CEO's breadth is significantly negatively related to firm performance, and the quadratic term is significantly positive. A CEO's depth is significantly positively related to firm performance, and the quadratic term is significantly negative. The research results indicate a u-shaped relationship between a CEO's breadth and firm performance and an inverted u-shaped relationship between their depth and firm performance. The study's findings extend the literature on factors influencing firm performance and CEOs' functional experience. The study expands from the horizontal macro to the vertical micro level, providing new evidence to support the recruitment and selection of high-level corporate talent.

## 1. Introduction

The effectiveness of chief executive officers (CEOs) as senior managers and their relationship with firm performance has been the focus of several strategic management studies. CEOs enjoy great discretion and freedom of action [1]. Compared to other executives, they have a more direct and essential effect on firm performance. According to the China Stock Market Accounting Research (CSMAR) database, CEOs' experiences are mixed. For example,

3). Click Home Page / Data Center / Single table query / Company Research series / Governance structure successively. Among them, is curricula vitae data from the "Executive Profile" data table. 4). Click Home Page / Data center / Single table query / Company research series / Basic information of listed companies successively. Among them, is the listed company data from the "listed company Basic Information" data table.

**Funding:** The authors received no specific funding for this work.

**Competing interests:** The authors have declared that no competing interests exist.

Huateng Ma, the CEO of Tencent, a listed company, has been working for Tencent since 1998, when he incorporated the Shenzhen Tencent Computer System Co. In September 2021, Tencent was selected as one of the Fortune 500 China list companies, ranking sixth. As another example, Zhang Yong, the CEO of the Alibaba Group, joined the company in August 2007 as a chief financial officer (CFO) and continued to serve as Alibaba's CEO until 2013. Before that, he worked as an auditor, department manager, and CFO at PricewaterhouseCoopers, Shanghai and Shanda Interactive Entertainment Ltd. On July 20, 2021, Alibaba Group ranked 18th on the Fortune 500 China list. From a simple statistical view, some representative companies, such as Tencent and Alibaba, have achieved excellent firm performance, but the experience of their CEOs is very different. Moreover, there are also other CEOs whose experience falls somewhere in between. From the CEOs' view, we were interested in the differences in their experience. At the same time, it is also worth paying attention to the direction of the firm performance results of these differences. The question arises: What characteristics of a CEO's experience are more conducive to improved firm performance? Examining this question may enable us to better understand the motivation for CEO selection and facilitate a reasonable and targeted CEO evaluation.

As early as 1984, scholars began to focus on the relationship between executive functional experience and firm performance. According to the upper echelons theory, Hambrick et al. argued that the characteristics of executives determine their experience, judgments, values, and leadership style. They also affect their perceptions of the strategic context in which the firm is placed and further influence its decisions and performance [1]. The upper-echelon theory has generated profound discussions and has been enriched over time. In addition to demographic backgrounds such as gender [2–4], age [4, 5], and education [6–8], executives' tenure and backgrounds, such as functional experience, will also have an impact on their decision-making behavior. This paper's purpose was to increase researchers' interest in continuing to study CEOs' functional experience characteristics and their effects on firm performance. Given this objective, this paper sought to advance the literature on CEOs' functional experience through two dimensions.

Functional experience is the experience of CEOs in various positions. Breadth refers to the broadness of their experience in various positions. The broader the variety of positions, the more beneficial it is to broaden the width of the knowledge base and increase the variety of knowledge sources. Depth refers to the average tenure, the hierarchy of position, and the depth of information or knowledge drawn from the knowledge sources during that tenure, which is conducive to weakening the risk and cost of knowledge transfer and reducing information asymmetry through deep functional experience. Scholars have dissected CEOs' characteristics from different perspectives, including four significant aspects of a CEO's duality (the two positions of board chairman and CEO in one), functional experience (single functional experience and integrated functional experience) [9–12], tenure [13], and political connections [14–18]. Since duality and functional experience focus on studies of the horizontal development of CEOs' positions, both studies are grouped into the literature covering breadth. In contrast, since tenure and political connections are more concerned with the vertical growth of CEOs in their positions, both studies are grouped into the literature about depth.

The relationship between CEO function experience and firm performance seems to be a black-box problem. Thus, how to further analyze this problem has motivated the researchers to carry out this study. Many studies support the idea that CEO experience affects firm performance. However, there are still some shortcomings in the existing research. (1) The breadth-based research does not consider the diversity and duplication of functions, and the rough count of the number of positions is not enough to reveal the breadth of the CEO's service experience. (2) The depth-based research does not consider the tenure of each stage of

employment, and these studies do not have a hierarchy of positions. (3) Current data analysis and processing methods are not always scalable, especially manual processing methods cannot automate research on a wide range of listed companies. Hence, these factors motivated the researchers to conduct this study.

Our study makes three contributions to the literature. First, compared with the subjective judgment of manual collation and questionnaires, this paper processed the data using cutting-edge information technology methods such as named entity recognition. We collected CEOs' resume data for 15 consecutive years from 2006 to 2020 to obtain comprehensive experience details. The analysis results are scientific and objective, and the technical approach can be extended to other types of executives. Second, compared with studies that mainly focus on the job category and tenure of executives' experience, this article introduced the breadth and depth dimensions of CEOs' functional experience enriching the study of the upper echelons theory. Finally, a few studies have provided partial evidence of the relationship between a CEO's experience and firm performance. In addition to the direct effect of the CEO's experience on firm performance, this paper further emphasized the nonlinear relationship between the two dimensions. The findings of this paper may serve as direct evidence for related studies to expand the literature on CEOs' experiences.

## 2. Literature review

Scholars have conducted in-depth studies of the factors influencing firm performance. In particular, research on CEOs' functional experience has achieved remarkable results. It is difficult to unify the conclusions of empirical analysis of different characteristics of firm performance. The research on CEOs' functional experience characteristics needs to explore the comprehensive and deep-seated mechanism of influence. This paper systematically analyzed the impact of CEOs' experience on firm performance from two perspectives: breadth and depth.

### 2.1 CEO's functional experience breadth and firm performance

In recent years, scholars have begun to pay attention to the breadth dimension of CEOs' functional experience. Studies of CEOs' duality and functional experience have both produced different perspectives [19–21]. The critical roles of the CEO and board chairman in a company are different. The CEO is responsible for managing and operating the company; the board chairman is responsible for leading the board. Duality occurs when the CEO also serves as the board chairman. Agency theory suggests that such duality creates a powerful CEO. They may use their power to hinder the board's decisions and, thus, affect the firm's performance [9, 12]. Management theory has argued that a CEO's duality establishes strong leadership through unified command. Therefore, CEOs may make better decisions that benefit firm performance [9, 12]. Other scholars have studied the effect of CEOs' functional experience on firm performance. Hamori et al. investigate the impact of prior CEO experience on firm performance [22]. They find that experience in the CEO position is negatively related to firm performance. Also, Georgakakis et al. provide evidence on the relationship between CEO characteristics and organizational performance [23]. Their research demonstrates that the performance implications of a new CEO origin should not be considered in isolation but in interaction with multi-level characteristics. Chahyadi et al. examine how the type of CEOs' industry experience (whether a CEO has cross-industry or specific-industry experience) on firm performance [24]. They find that CEOs with cross-industry experience tend to relatively lower firm performance. On the other hand, CEOs with specific-industry experience lead firms to higher performance, especially among high-growth firms. Zhang et al. showed that executives who have returned from working overseas could improve firm performance, especially in non-state-owned

enterprises, large firms, and firms in East China. Further analysis of this mechanism shows that overseas-returned executives influence firm performance mainly through risk-taking mechanisms [25]. The results of [26] also supported the contention that firms experiencing low performance were more likely to recruit a CEO with a background in operations.

In addition to single functional experience studies, some scholars have also conducted studies of multiple functional experiences. Frederiksen et al. found a significant positive relationship between the number of roles experienced by a senior person in the labor market and their chances of career success [27]. A series of studies of different functional experiences were generated based on this. On the one hand, studies based on the domain expertise framework [27] argued that a CEO's combined functional experience could leverage specific resources. These experiences can enhance adaptability [28], improve the firm's relationship with investors [28], and help the firm to escape from financial difficulties [29] and transfer across companies [30], thus, contributing to the improvement of the firm performance. On the other hand, CEOs with combined functional experience face ambiguity when adjusting the external environment and the current resources available to the company, which may lead to lower firm performance, as the combination of experience with firm-specific resources achieves limited success when driven by different achievements [31]. In addition, Li and Patel investigate the impact of CEO generalist experience and firm performance [32]. They propose a negative association between more generalist CEO experience and firm performance, but one that is alleviated by longer tenure. In particular, knowledge and industry are less relevant in new environments. Based on these factors, the combined functional experience of CEOs may be negatively related to firm performance.

## 2.2 CEO's functional experience depth and firm performance

Studies of the depth of CEOs' functional experience have focused on the length of tenure and the hierarchical structure of political connections. Tenure-based studies mainly consider the effective accumulation of relevant expertise gained by managers as tenure increases, including the ability to be more thoughtful in strategy formulation and implementation [33], the ability to achieve implicit learning through a better understanding of practices and existing knowledge [34], and the ability to learn experientially through a combination of resources [35]. All three modes may enable firms to achieve better performance returns. Resource-based theory research suggests that a CEO accumulates a wealth of knowledge and organizational memory in the field during a long tenure and that these contribute to the firm's performance [36]. Bergh et al. showed that a CEO's tenure is positively related to firm performance [36, 37], and this finding is empirically supported by other researchers [38, 39]. Studies based on the upper echelons theory suggest that longer tenure is detrimental to firm performance. Hambrick et al. showed that long-tenured CEOs often rely on internal channels and ego models to achieve success [13]. They become risk-averse and increasingly distant from fresh and accurate external market information [36, 40]. Therefore, this is detrimental to firm performance [41]. Cai et al. examined the relationship between a CEO's tenure and firm performance. Their findings also indicated that a longer CEO tenure positively affects firm performance.

In addition to tenure, scholars have made a series of explorations regarding the depth of functional experience through the hierarchical structure of CEOs' political connections. Almost all executives are politically connected [41]. Resource dependence theory suggests that CEOs with political affiliations help firms obtain resources to cope better with challenging contingencies. Political connections may, thus, benefit firm performance. Wu et al. suggested that CEOs' political connections positively impact firm performance. This impact is stronger in less-developed regions [42]. In contrast, Ling et al. used a sample of real estate firms. They

showed that political connections are negatively related to firms' financial performance. Cultivating political connections may be risky and not always reward performance [43].

In summary, existing research demonstrates the centrality of CEOs' functional experience in studying firm performance. However, there are some shortcomings in the existing studies. Breadth-based studies do not consider the diversity and repetition of functions. An approximate count of the number of positions is not sufficient to reveal the breadth of CEOs' experience. Depth-based studies do not consider the length of tenure at each stage, and these studies do not contain a hierarchy of individual positions. Therefore, this paper combined breadth and depth to portray CEOs' cumulative perceptions and investigate whether CEOs' functional experience can affect firm performance.

## 3. Theory and Hypothesis

### 3.1 CEO's functional experience breadth and firm performance

Hambrick and Mason first proposed the upper echelons theory in 1984. This theory links the demographic characteristics of executives to firm outcomes and argues that executive characteristics are ultimately reflected in corporate strategy choices and firm performance [43]. Executives play a dominant role in firm management. Managers may have different options when faced with corporate decisions because of the potential influence of their characteristics.

The upper echelons theory suggests that in complex corporate environments, individual executives' decisions are driven by "limited rationality" in the face of complex and uncertain situations [1]. Recent studies have found that CEO's experience hinders subsequent firm performance [22, 44]. The rationale is that prior experience is too heavily laden with the specific environments in which it was gained and therefore is not as beneficial to the new firms as the CEOs believe it will be. In short, available evidence suggests that past experience is a detriment to success in a subsequent job, mainly because experienced CEOs have to give up firm-specific skills that are not useful to a new firm [22]. It is exacerbated by the tendency of CEOs to become less adaptable with increasing experience [40, 45, 46]. Therefore, as the CEO's functional experience increases, they face more and more instability in the environment and decision-making, which is not conducive to improving the firm performance [47]. The upper echelons theory believes that the individual's experience characteristics also affect risk preference [1]. When a listed company hires a CEO, their experience is usually considered. In reality, more and more listed companies hire experienced CEOs because they can provide more professional policy advice for the company's development. On the one hand, experienced CEOs can apply their expertise to practical decisions, helping to make sound decisions. On the other hand, reliable and prudent practice requirements will also prompt them to form corresponding characters and habits with a lower risk appetite [48]. Suppose the incentive mechanism is not complete and the supervision system is not perfect, CEOs may make the pursuit of maximizing their interests their goal, borrowing managerial authority and information advantage for their development to gain more benefits, regardless of improving company performance.

However, improved firm performance is associated with market share expansion and technological innovation. High firm performance may be accompanied by more significant risk. With sufficient functional experience, the CEO has ample time and opportunity to bond with the company's staff and mechanisms to mitigate the company's nurturing costs. After the CEO and the other members have connected, exceptional productivity is created, in addition to a working understanding, showing a "special cooperation" relationship. Furthermore, the wealth of experience enables them to replicate success stories. It is used to resist customer churn, technological conservatism, and declining performance, ensuring a steady improvement in company performance [1].

Based on the above analysis, firm performance weakens with increased CEO richness. However, with further expansion of functional experience richness, human capital and risk resistance are strengthened, enhancing firm performance. As a result, the following hypothesis was proposed.

**H1**: *As a CEO's functional experience breadth increases, it has a u-shaped relationship with firm performance.*

## 3.2 CEO's functional experience depth and firm performance

As a firm's business grows, the scope and complexity of its CEO's unconventional strategic decisions become more extensive. However, most CEOs lack the necessary information-processing skills to deal with unexpected events [49]. As the depth of experience increases, the CEO accumulates sufficient resources from higher levels and longer tenure beyond the current company. It is easier for them to make strategic decisions to improve firm performance. This view is supported by scholars who argue that CEOs' knowledge of the growth of their tenure is conducive to enhancing organizational competitiveness and firm performance. At the same time, the market law of survival of the fittest means that CEOs with poor performance are eliminated early. Those with greater in-depth experience are relatively more capable talents. CEOs with deep experience mark personal ability and better firm performance [50]. It has also been shown that CEO promotion positively affects firm performance [51]. Greater career opportunities improve CEOs' managerial motivation. The career concern theory suggests that CEOs have two sources of reward for their efforts. One is the monetary reward of the current job, and the other is the career reward of the current job [51]. These two components together motivate CEO effort, and in the context of this paper, the expectation was that a CEO's depth of tenure would result in higher returns for firm performance.

However, as the depth of experience increases, the CEO's aversion to risk increases because they represent a significant investment in the company's future, both on a psychological level and in terms of actual commitment [40]. Risk aversion leads CEOs to resist changes in the status quo, which may be detrimental to firm performance. On the other hand, there is an increasing reliance on internal resources. As these resources have often learned how to cater to the CEO's information preferences, it isn't easy to accept new information [13]. At the same time, due to limited rationality constraints [1], CEOs are less likely to consistently address the information and knowledge inputs required for firm performance [13]. In particular, as the depth of a CEO's tenure increases, they are prone to directional thinking, progressively more information asymmetry, and reduced information processing power. As demonstrated in Kale et al.'s study, CEOs are at the top of the firm for long periods and lack promotion-based incentives [52]. Second, as their tenure increases, overconfidence from existing achievements increases. They may blindly maintain their own decisions, reducing the company and industry fit. Furthermore, if they remain in the same hierarchy for a long time, they are limited by the information sources window. The content of the information they are exposed to and receive becomes increasingly monotonous and is not conducive to their firm's performance improvement.

Based on the above analysis, firm performance is enhanced with the depth of a CEO's tenure. However, as this depth deepens, information asymmetry and solidification of thinking bring increasing numbers of problems, and the focus of attention changes, which may hurt firm performance. As a result, the following research hypothesis was proposed.

**H2**: *As a CEO's functional experience depth increases, it has an inverted u-shaped relationship with firm performance.*

## 4. Experimental design

### 4.1 Data Source

The sample data of this paper includes all listed companies in China between 2006 and 2020. This paper mainly focused on CEOs of listed companies due to the following considerations. First, the information disclosure requirements of listed companies in China ensure the availability and reliability of data. Second, CEOs of listed companies have a certain degree of mobility, reflecting the mobility of China's managerial market to a certain extent. The study can provide a preliminary understanding of the current situation of CEOs, which can then be extended to the study of other types of executives. Third, this paper is based on the upper echelons theory. The CEOs' organizational environment and industry experience of China's listed companies are relatively wealthy.

The Measures for the Administration of Information Disclosure of Listed Companies was considered and adopted by the China Securities Regulatory Commission in 2006. The data disclosure is complete, so the research data in this paper starts in 2006. This paper automatically extracted data related to functional experience from CEOs' CVs based on the deep learning model for CEOs' functional experience richness. Among the data, the resume information was obtained from the CSMAR database. However, there was a problem in that some resume information was missing or hidden. Building a targeted training corpus and maximizing the model's performance was necessary to obtain more comprehensive and accurate information about CEOs' tenure. Therefore, this paper also used other information channels to supplement executive CVs. All the CVs obtained from different channels were cleaned using data fusion, and finally, a relatively complete resume text was obtained.

This paper used two variables, return on total assets (ROA) and return on equity (ROE), to measure firm performance. Data for other corporate governance and control variables, such as financial indicators, were obtained from the CSMAR and Wind databases. The initial sample was processed as follows: (1) sample companies from the financial industry were excluded; (2) sample companies with missing data were excluded; (3) to reduce the influence of individual extreme values, all continuous variables were subjected to tailoring at the 1% level. Finally, this paper obtained 15 years of data from 1,862 companies with 27,930 samples. Table 1 shows the variable definitions, and the descriptive statistics of the variables after the tailoring process are shown in Table 2.

### 4.2 Variable design

Firm performance: This paper followed the common practice in the literature to construct firm performance indicators. This refers to the performance and efficiency of a firm's operations, which reflects their effectiveness and is generally reflected by a financial indicator or a set of financial indicators. Typical measures of firm performance in the literature are based on accounting-level calculations, commonly represented by measures such as ROA and ROE [53, 54]. Therefore, in this paper, ROA and ROE represent firm performance.

A CEO's functional experience: This paper analyzed the functional experience of CEOs in each period. It can be seen from Fig 1 that most CEOs have undergone the evolution of different hierarchies of positions before reaching the highest position. With the support of junior levels such as workers, station managers, and workshop technicians, they enter the core circle of senior-level positions such as directors, managers, and boards of directors through the experience of intermediate-level positions like higher-level project managers, section chiefs, and directors.

Breadth: **func_rich** was constructed by the functional experience richness of the CEO. This paper used the information entropy formula to calculate the functional experience richness, as

**Table 1. Variable definition.**

| Type | Name | Description |
|---|---|---|
| **Dependent variable** | ROA | Return on Assets, i.e., the ratio of net income and total assets |
| | ROE | Return on Equity, i.e., the ratio of net income to shareholder's equity |
| **Independent variable** | func_rich | See Eq (1) for more detail |
| | level_score | See Eq (2) for more detail |
| | avg_emp | The average tenure of each position |
| **Control variables** | gender | Males take 1; females take 0 |
| | age | The corresponding age in the statistical year |
| | degree | 1 denotes a degree below junior college; 2 denotes a junior college degree; 3 denotes a bachelor's degree; 4 denotes a master's degree; 5 denotes a doctoral degree. |
| | sharend | Number of shares held by CEO (larger value, for the sake of data balance, the unit is taken as million shares) |
| | dar | Total liabilities/total assets |
| | size | Natural logarithm of the total assets of the company |
| | nstaff | Number of all employees of the company (the value is large, for the sake of data balance, the unit is a thousand) |
| | found_time | Time from company inception to the cut-off date |
| | exe_num | Number of the senior management team |

shown in Eq (1). Where $p_i$ is the frequency of the $i$th category of tenure position in the career, and the value range of $i$ is [7, 50]. Positions in the resume were divided into nine categories: production, R&D, design, human resources, management, marketing, finance, legal, and others. For example, if a CEO held 13 positions in their career, of which three were in the finance category, the frequency of positions in the finance category is 3/13. The higher the final calculated score, the more experience the CEO has in each category of positions and the broader the CEO's breadth.

$$func\_rich = -\sum p_i \log(p_i) \tag{1}$$

Depth: A CEO's functional experience depth was measured by the function hierarchy

**Table 2. Descriptive statistics.**

| VarName | Obs | Mean | SD | Min | Max |
|---|---|---|---|---|---|
| roa | 27930 | 0.0452 | 0.0552 | 0 | 0.2690 |
| roe | 27930 | 0.0691 | 0.0691 | 0.1445 | 0.4499 |
| func_rich | 27930 | 0.2948 | 0.7812 | 0.1727 | 0.9030 |
| level_score | 27930 | 1.0631 | 0.3771 | 0.2320 | 2.0049 |
| avg_emp | 27930 | 3.8614 | 1.2066 | 0.3333 | 18.6667 |
| gender | 27930 | 0.9354 | 0.2458 | 0 | 1 |
| age | 27930 | 49.2939 | 1.8176 | 29 | 62 |
| degree | 27930 | 3.5364 | 1.3305 | 1 | 5 |
| sharend | 27930 | 287.7353 | 6.0153 | 0 | 3802.6100 |
| dar | 27930 | 0.4196 | 0.4196 | 0.0208 | 0.9803 |
| size | 27930 | 9.7258 | 9.7258 | 8.3840 | 13.2253 |
| nstaff | 27930 | 3.0895 | 3.0895 | 1.8260 | 5.3206 |
| exe_num | 27930 | 19.8121 | 4.9925 | 9 | 52 |
| found_time | 27930 | 8.1397 | 6.2352 | 0 | 25 |

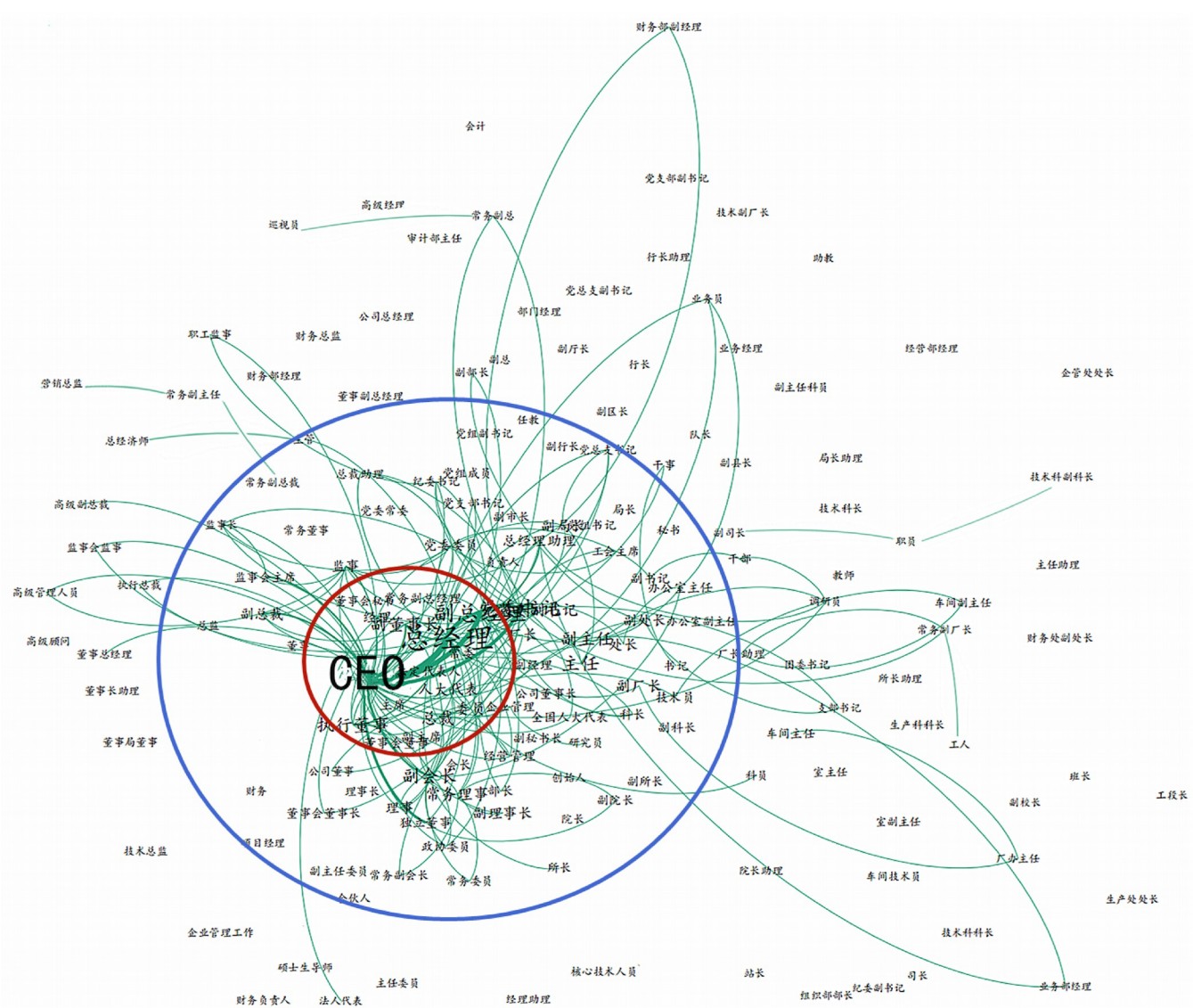

**Fig 1. Evolution of CEO's position promotion chart.**

(*level_score*) and the average tenure of each position (*avg_emp*). Where $p_i$ is the ith category of tenure frequency in the career, $r_i$ is the number of times the ith tenure hierarchy appears, and $a_i$ is the weight of ith level score. The levels were divided into senior, middle, and junior. Combined with the structural ratio of senior, intermediate, and junior positions [55], this paper gave a weighted score of 6:3:1 to senior, intermediate, and junior levels.

On August 31, 2006, the Ministry of Human Resources and Social Security of the People's Republic of China issued the Implementation Opinions of the Tentative Measures for the Management of Post Establishment in Public Institutions. According to Article 13 of the setting of professional and technical post levels states: the structural proportion of senior, intermediate, and junior posts shall be controlled according to the regional economic and social development level and industrial characteristics, as well as the function, specification, subordination and professional and technical level of the public institution. The overall national control target is 1:3:6 for the proportion of senior, intermediate, and junior posts. Combined with

the structural ratio of senior, intermediate, and junior positions, this paper gave a weighted score of 6:3:1 to senior, intermediate, and junior levels.

$$leveLscore = -\sum \alpha_i r_i \tag{2}$$

**4.2.1 Control variables.** This paper referred to the variables used in the firm performance research literature and added variables that may affect firm performance, such as the number of shares held at the end of the year, the time the company was listed, and the number of members of the executive team, which are shown in Table 2. According to the upper echelons theory, the demographic characteristics of executives have an impact on firm performance. This paper also controlled typical personal characteristics such as *gender*, *age*, and *education*. Studies have shown that higher levels of CEO shareholding are conducive to improved firm performance, so this paper controlled the number of shares held by the CEOs. In addition to the CEO's level, this paper also controlled for the firm-level variables: *dar* is the gearing ratio; the higher the gearing ratio, the greater the financial risk faced by the firm, and the resulting debt service pressure conflicts with the continuous need for cash flow for firm performance improvement [56, 57]. The expression *size* is the firm's scale, expressed as the natural logarithm of total assets at the end of the period. The larger the firm size, the more resources are available to carry out activities to improve the firm's performance [58, 59]. The expressions *exe_num* and *nstaff* are the number of members of the firm's executive team and all employees, respectively [60, 61].

The CEO does not operate in a vacuum; they decide upon and check the strategic decisions made by the executive team members to improve the firm's performance. Hence, they play a controlling role. Therefore, this paper controlled the number of members of the executive team and the number of employees of the firm. The expression *found_time* is the time since the company was founded [62]. With the growth of the length of time since the company's founding, some studies have shown that the company's performance has steadily increased. However, other studies have shown that the company's aging leads to a rigid organizational management model, which is not conducive to improving the company's performance. The company's founding time was included within the control variables to explore better the characteristic variables of the CEO's functional experience.

Table 2 shows the descriptive statistical analysis of this paper. The data shows, in terms of firm performance, that the average ROA of the firms in the sample period is 0.0452. The minimum value is 0, and the maximum value is 0.2690. This trend indicates that companies' performance level varies widely due to industry and competitive market environment factors. In terms of the three indicators of CEOs' functional experience, the range of values for the breadth indicator is from 0.1727 to 0.9030, which indicates that the richness of tenure among CEOs is highly variable; the two indicators, including depth, both in terms of depth and the time spent in each tenure, are relatively different. Taking the average time in office as an example, the shortest is 0.33 years, and the longest is 18.67 years. This shows that some CEOs left after about four months in a position, while some CEOs had held onto a position for 18 years. CEOs' experience is diversified and enriched at the initial data-driven level, and the wide variation in experience among CEOs warrants in-depth exploration.

## 4.3 Model design

According to the characteristics of panel data, we test the validity of the fixed-effects model and the random-effects model on the validity of the regression model. Usually, the Hausman test is used to judge the applicability of the two models. The p-value of the Hausman test result

is 0.0000, so the null hypothesis is strongly rejected, and a fixed-effects model should be used instead of a random-effects model.

This paper adopts the two-way fixed effects model. *Yeardummy* and *Industrydummy* represent the year dummy and industry dummy variables, respectively. Both of them are 0–1 dummy variables. $\mu_{it}$ is the model error term. i is the ith firm. t is the t year. The above control and dummy variables are added to the model to control the influence of other factors further. In this paper, two models were designed to test the hypotheses. The first model (1) is shown in Eq (3), and the second model (2) is shown in Eq (4). Model (1) tests the direct effect of a CEO's breadth of experience and depth of experience on firm performance. Model (2) tests the relationship between a CEO's breadth of functional experience and the depth of functional experience and firm performance. In addition to ROA, this paper also examined ROE as a measure of firm performance, replacing ROA with ROE and keeping the other variables the same as these two models.

$$\begin{aligned}
\mathrm{ROA}_{it} = {}&\propto_0 + \propto_1 \mathrm{func\_rich}_{it} + \propto_2 \mathrm{level\_score}_{it} + \propto_3 \mathrm{avg\_emp}_{it} + \propto_4 \mathrm{age}_{it} + \propto_5 \mathrm{degree}_{it} \\
&+ \propto_6 \mathrm{sharend}_{it} + \propto_7 \mathrm{dar}_{it} + \propto_8 \mathrm{size}_{it} + \propto_9 \mathrm{nstaff}_{it} + \propto_{10} \mathrm{found\_time}_{it} \\
&+ \propto_{11} \mathrm{exe\_num}_{it} + \mathrm{Yeardummy} + \mathrm{Industrydummy} + \mu_{it} (3)
\end{aligned}$$

$$\begin{aligned}
\mathrm{ROA}_{it} = {}&\beta_0 + \beta_1 \mathrm{func\_rich}_{it} + \beta_2 \mathrm{level\_score}_{it} + \beta_3 \mathrm{avg\_emp}_{it} + \beta_4 \mathrm{func\_rich}_{it}^2 + \beta_5 \mathrm{level\_score}_{it}^2 \\
&+ \beta_6 \mathrm{avg\_emp}_{it}^2 + \beta_7 \mathrm{age}_{it} + \beta_8 \mathrm{degree}_{it} + \beta_9 \mathrm{sharend}_{it} + \beta_{10} \mathrm{dar}_{it} + \beta_{11} \mathrm{size}_{it} \\
&+ \beta_{12} \mathrm{nstaff}_{it} + \beta_{13} \mathrm{found\_time}_{it} + \beta_{14} \mathrm{exe\_num}_{it} + \mathrm{Yeardummy} + \mathrm{Industrydummy} \\
&+ \varepsilon_{it} (4)
\end{aligned}$$

## 5. Empirical analysis

### 5.1 CEO's breadth, depth, and firm performance

The regression results of model (1) are shown in column (1) in Table 3. The regression results of model (2) in column (3) are in Table 3. Both are the results of the impact of a CEO's breadth of tenure and depth of tenure on firm performance in the case of firm performance expressed by ROA.

At the breadth level, the regression results in column (1) show that the regression coefficient of *func_rich* is significantly negative at the 1% level. The regression results in column (3) show that the regression coefficient of *func_rich* is significantly negative at the 1% level, and the regression coefficient of its quadratic term *func_rich2* is significantly positive at the 5% level. Therefore, there is a u-shaped change in the relationship between a CEO's functional experience richness and firm performance as it increases, and hypothesis one of this paper is verified.

At the depth level, the regression results in column (1) show that the regression coefficients of *level_score* and *avg_emp* are both significantly positive at the 1% level. The regression results in column (3) show that *level_score* and *avg_emp* are significantly positive at the 1% level. The regression coefficients of their quadratic terms are significantly negative at the 5% level. There is an inverted u-shaped change in the relationship between a CEO's depth and firm performance, increasing from weak to strong. Hypothesis two of this paper is verified.

Similarly, columns (2) and (4) both show the results of the effect of a CEO's breadth of tenure and depth of tenure on firm performance when firm performance is expressed in terms of ROE. The regression results also support the two hypotheses of this paper.

The regression coefficient of the CEO's *gender* is significantly positive, which indicates that male CEOs are better at promoting firm performance. The regression coefficient of CEOs' education (*degree*) is significantly positive, indicating that CEOs accumulate sufficient

**Table 3. Regression results.**

| | 1 | 2 | 3 | 4 | 5 | 6 | 7 |
|---|---|---|---|---|---|---|---|
| | ROA | ROE | ROA | ROE | TobinQ | PEF | ROA |
| func_rich | -0.1539*** | -0.0675*** | -0.2508*** | -0.3026* | -0.6005* | -0.8921* | -0.1086* |
| | (-6.374) | (-3.5437) | (-6.1475) | (-1.8644) | (-1.6917) | (-1.7629) | (-1.9034) |
| level_score | 0.0273*** | 0.0310*** | 0.1195** | 0.0618** | 0.4304* | 0.5462* | 0.0625** |
| | (3.3571) | (2.9673) | (1.9896) | (1.9735) | (1.8493) | (1.7383) | (2.0365) |
| avg_emp | 0.3571*** | 0.2436** | 0.0837*** | 0.0137** | 0.251*** | 0.071*** | 0.1068* |
| | (7.7514) | (2.0235) | (10.2567) | (2.2366) | (2.9658) | (2.3852) | (1.6950) |
| func_rich2 | | | 0.0195** | 0.0453*** | 1.3531*** | 2.4396*** | 0.1562** |
| | | | (2.4148) | (4.1089) | (14.2755) | (5.2445) | (2.1502) |
| level_score2 | | | -0.0150** | -0.0094** | -0.0218*** | -0.1270*** | -0.0369* |
| | | | (-1.9846) | (-2.1375) | (-3.4436) | (-4.0034) | (-1.7561) |
| avg_emp2 | | | -0.0112** | -0.0371*** | -0.0294* | -0.0277* | -0.6524** |
| | | | (-2.4946) | (-2.6744) | (-1.7435) | (-1.7177) | (-1.9806) |
| gender | 0.0345** | 0.2405* | 0.0253** | 0.0046** | 0.0247** | 0.0386* | 0.0207*** |
| | (2.3751) | (1.9739) | (2.0693) | (2.3509) | (2.1473) | (1.6704) | (10.2648) |
| age | -0.0243** | -0.0326** | -0.0125** | -0.0972** | -0.9850*** | -0.9637*** | -0.2394*** |
| | (-2.0437) | (-2.1550) | (-2.4105) | (-2.2433) | (-10.4502) | (-11.6924) | (-6.2341) |
| degree | 0.0278** | 0.0994** | 0.0488** | 0.0655** | 0.0426** | 0.1297*** | 0.6512** |
| | (2.4368) | (2.1466) | (1.9906) | (1.9842) | (2.1799) | (4.1673) | (2.2359) |
| sharend | 0.0037** | 0.0140** | 0.0097* | 0.0107** | 0.6701* | 0.1652 | 0.5309** |
| | (2.1937) | (2.4467) | (1.7326) | (1.9938) | (1.7796) | (0.3599) | (2.0215) |
| dar | -0.1375*** | -0.1769* | -0.1030*** | -0.0639*** | -0.0771*** | -0.1074*** | -0.2241* |
| | (-12.342) | (-1.7388) | (-10.6322) | (-6.3781) | (-5.3549) | (-7.6017) | (-1.8503) |
| size | 0.0457** | 0.3470** | 0.0286*** | 0.0370*** | 0.0459** | 0.0524** | 0.0356** |
| | (2.0378) | (1.9835) | (5.1301) | (2.7364) | (2.8427) | (2.9004) | (2.3688) |
| nstaff | 0.0137* | 0.1738* | 0.0093* | 0.0136* | 0.2786*** | 0.2769*** | 0.3640** |
| | (1.8633) | (1.7026) | (1.6937) | (1.7037) | (0.2756) | (6.6216) | (1.9931) |
| exe_num | 0.0375** | 0.3786* | 0.0235** | 0.0969** | 0.1540* | -0.0854* | 0.5625* |
| | (1.7556) | (1.8349) | (1.9907) | (2.4377) | (1.9404) | (-1.6854) | (1.9002) |
| found_time | 0.0004* | 0.0237* | 0.0015* | 0.0059** | 0.0623*** | 0.0081*** | 0.0035** |
| | (1.9043) | (1.7095) | (1.7015) | (2.1831) | (6.8297) | (5.8853) | (2.3157) |
| _cons | -1.0655 | -1.7234 | -9.0008 | -9.7231 | -5.0042 | -3.0532 | -7.2315 |
| | (-1.0373) | (-0.6085) | (-0.7713) | (-0.5874) | (-0.0136) | (-0.5201) | (-0.2679) |
| Industry-year fixed effects | Yes | Yes | Yes | Yes | Yes | Yes | Yes |
| N | 27930 | 27930 | 27930 | 27930 | 27930 | 27930 | 4530 |
| R-sq | 0.7758 | 0.7729 | 0.7648 | 0.7905 | 0.8742 | 0.8527 | 0.8865 |
| adj. R-sq | 0.7145 | 0.7426 | 0.7457 | 0.7358 | 0.8172 | 0.8305 | 0.8631 |

knowledge with their increase in education, which benefits the company's performance. The regression coefficient of CEOs' **sharend** is significantly positive, indicating that the higher the level of a CEO's shareholding, the better the company's performance. Second, at the firm level, the regression coefficient of firm debt (**dar**) is significantly negative, indicating that the higher the firm's gearing, the greater the financial risk it faces, and the resulting proxy for debt service pressure conflicts with the continued need for cash flow for firm performance. The regression coefficient of firm **size** is significantly positive, indicating that the larger the firm is, the more resources it has to carry out performance improvement activities. The regression coefficients of the number of employees (**nstaff**) and executive team members (**exe_num**) are significantly

positive. The higher the number of team members, the more information is available. The CEO can obtain timely information, such as technical signals related to company performance, through bottom-up information gathering. The regression coefficient of **found_time** is significantly positive, indicating that the company accumulates greater resources as time increases, which is conducive to improving firm performance.

## 5.2 Robustness tests

**5.2.1 Alternative approach to firm performance variables.** Firm performance studies often use *ROA*, *ROE*, and *TobinQ* to represent firm performance. Among them, ROA and ROE are described in this paper's experiments, and TobinQ is the ratio of the market value of the firm to total assets. We use TobinQ measure of firm performance in the robustness test as our dependent variable. TobinQ ratio (or Q as it is commonly known), defined as the capital market value of the firm divided by the replacement value of its assets incorporates a market measure of firm value which is forward-looking, risk-adjusted, and less susceptible to changes in accounting practices [63, 64]. In order to better measure a company's performance, this paper used PCA (Principal Component Analysis) to analyze the three indicators of *ROA*, *ROE*, and *TobinQ*. PCA is to try to recombine the original many related indicators (such as P indicators) into a new set of comprehensive indicators independent of each other to replace the original ones. The results of the principal component analysis are shown in Table 4. Among them, (Component) represents the name of the principal component extracted by the system. It can be found that a total of 5 principal components have been extracted. The Eigenvalue column represents the eigenvalue of the principal component extracted by the system. The size of the eigenvalue means the explanatory power of the principal component. The larger the eigenvalue, the stronger the explanatory power. The Proportion column represents the variance contribution rate of the principal component extracted by the system, and the variance contribution rate is also the explanatory power of the principal component.

Ultimately, the characteristic root of the first principal component is 2.4820, which is greater than 1. The first principal component explains 82.73% of the total variables. The expression of the composite indicator of firm performance (*PEF*) obtained by expressing the first principal component as a linear combination of the individual variables is:

$$PEF = 0.5861ROA + 0.6049ROE + 0.5392TobinQ \tag{5}$$

The empirical test was conducted with *TobinQ* and *PEF* as the dependent variable. As shown in columns (5) and (6) in Table 3, the regression results support hypotheses one and two.

**Table 4. Regression results.**

| Principal components/correlation | | | | |
|---|---|---|---|---|
| **Component** | **Eigenvalue** | **Difference** | **Proportion** | **Cumulative** |
| Comp1 | 2.4820 | 2.0799 | 0.8273 | 0.8273 |
| Comp2 | 0.4021 | 0.2862 | 0.1340 | 0.9614 |
| Comp3 | 0.1158 | 0.0001 | 0.0386 | 1 |
| Principal components (eigenvectors) | | | | |
| **Variable** | **Comp1** | **Comp2** | **Comp3** | **Unexplained** |
| ROA | 0.5861 | -0.4998 | 0.6378 | 0 |
| ROE | 0.6049 | -0.2539 | -0.7548 | 0 |
| TobinQ | 0.5392 | 0.8281 | 0.1535 | 0 |

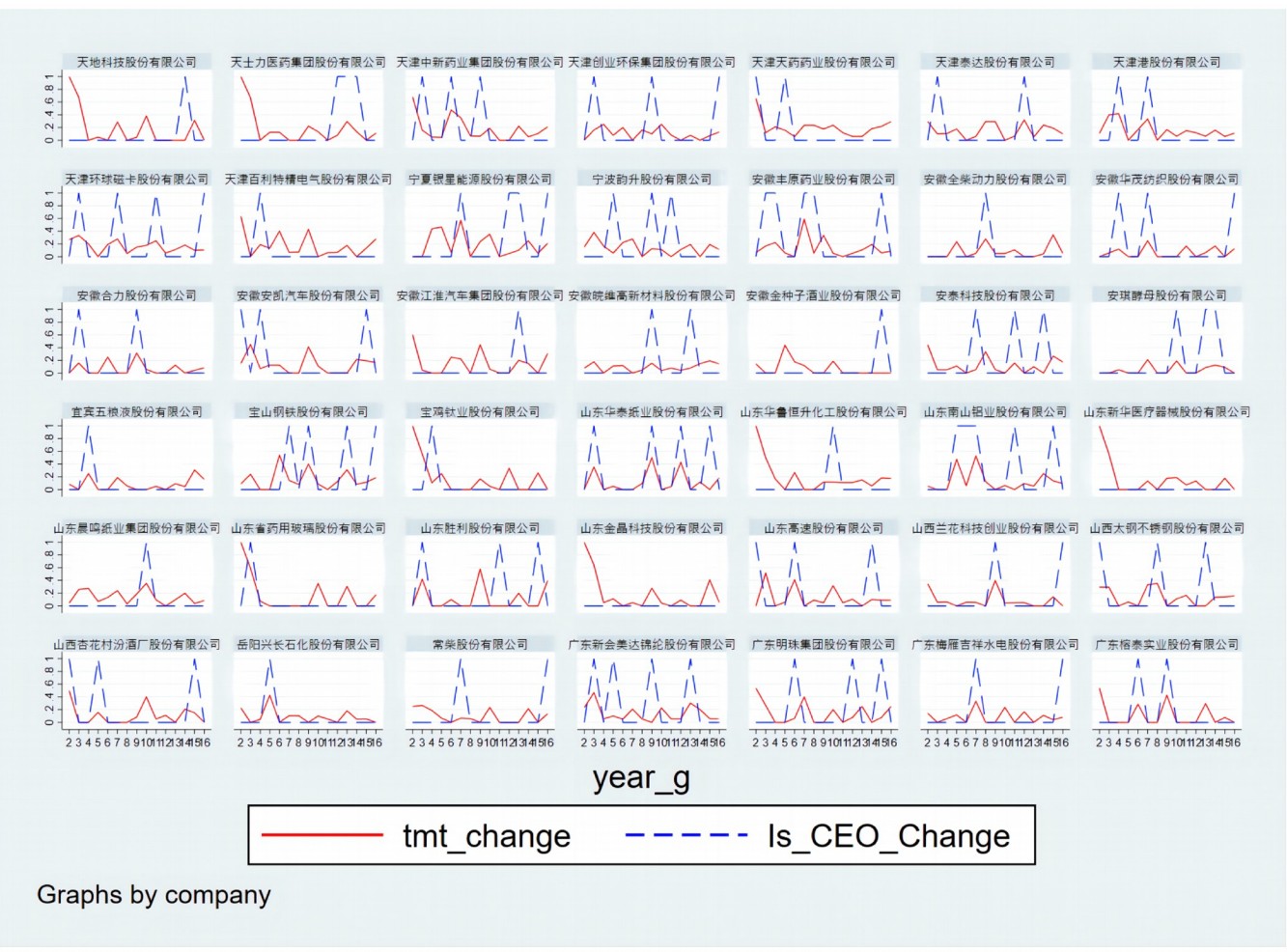

**Fig 2. The change rate of the CEO and top management team.**

**5.2.2 Control CEO endogeneity.** A potential problem with our findings is that listed companies have other executives besides the CEO. According to our statistics, the top management team changes almost every year. If the top management team is highly fluid, it is difficult to measure whether the CEO or other executives bring about an improvement in firm performance.

Therefore, this paper analysis the stability of the CEO and top management team each year. Fig 2 shows the CEO and top management team changes over the statistical period as a group of company representatives. The blue dotted line indicates whether the CEO has changed in the current year, 1 if the CEO changes, and 0 if there is no change. The solid red line represents the rate of change in the top management team.

As can be seen from the figure, most listed companies have changed their CEOs during the statistical period. The changes in the top management team are different. Some companies have undergone substantial changes in the top management team during the statistical period, while the changes were relatively small in others. We further make a descriptive statistic of the change rate of the top management team. The rate of change of the top management team is distributed between [0, 1]. The average rate was 13%. Therefore, this paper believes that if the current year's change rate of the top management team is less than 13%, it is relatively stable.

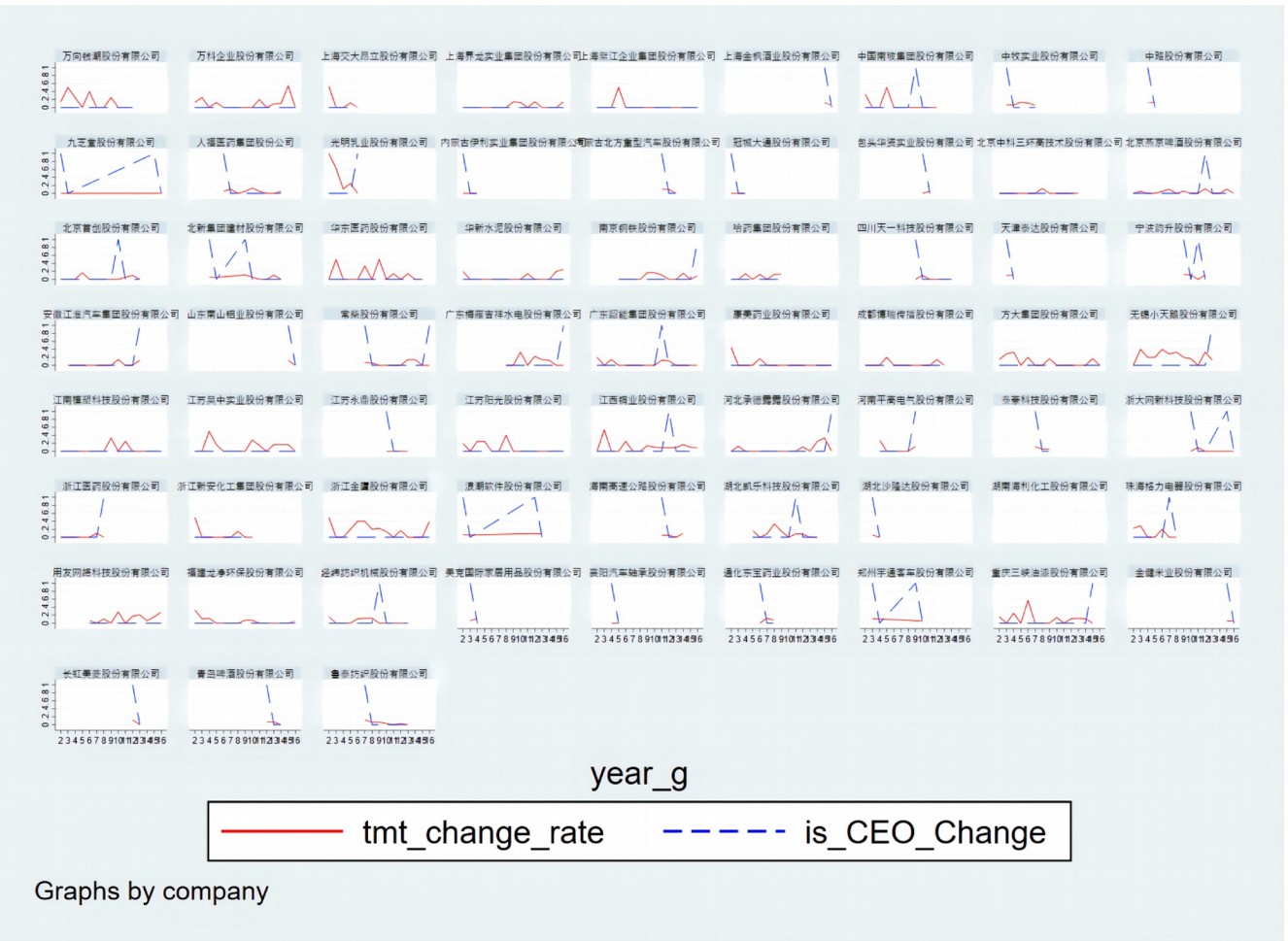

**Fig 3. The change rate of the CEO and Top management team in the filtered samples.**

According to this standard, this paper filtered the data of CEO changes and a relatively stable top management team for model testing. The part of the sample is shown in Fig 3.

The filtered samples are smaller but cleanest. In these samples, it is guaranteed that the CEO changes, but the top management team remains relatively stable. This method can better describe the relationship between CEOs' functional experience and company performance. The model empirical analysis results for this sample are in column 7 in Table 3. The model results still support Hypotheses 1 and 2.

Second, Instrumental Variable Analysis. Although the fixed effect model can solve the problem of missing variables to a certain extent. There may still be two-way causal problems. Without verification, we cannot rule out the possibility of a two-way causal relationship between firm performance and the CEO's functional experience. For example, due to the improvement of the firm's performance, it is possible to achieve a leading position in the whole industry or even more extensive cross-industry cooperation. As a result, CEOs have more opportunities for cross-border cooperation, thus enriching their functional experience. Of course, the above inference is only a possible guess. As for the possible two-way causal problem, this paper uses the instrumental variable method to test it further. An appropriate instrumental variable needs to meet two basic requirements: it is related to an endogenous

**Table 5. Regression results of instrumental variable.**

|  | 1 | 2 |
|---|---|---|
|  | **First step** | **Second step** |
|  | **func_rich** | **ROA** |
| func_rich | - | -0.0735*** |
|  | - | (-3.5437) |
| func_mean | 0.0802*** | - |
|  | -2.9601 | - |
| level_score | 0.0156*** | 0.0227*** |
|  | -2.9657 | -2.9946 |
| avg_emp | 0.2382*** | 0.1506** |
|  | -6.2684 | -2.1463 |
| gender | 0.0259** | 0.3501* |
|  | -2.4202 | -1.8561 |
| age | -0.0056** | -0.0651** |
|  | (-2.2341) | (-2.2304) |
| degree | 0.0659** | 0.0503** |
|  | -2.0637 | -2.0476 |
| sharend | 0.0135** | 0.0427** |
|  | -2.3407 | -2.0759 |
| dar | -0.2350*** | -0.3540* |
|  | (-7.5404) | (-1.8364) |
| size | 0.0240** | 0.2375** |
|  | -2.3541 | -2.2434 |
| nstaff | 0.1437* | 0.2413* |
|  | -1.7213 | -1.9834 |
| exe_num | 0.0357** | 0.3354* |
|  | -1.0254 | -1.9834 |
| found_time | 0.0127* | 0.4721** |
|  | -1.6923 | 2.0427 |
| _cons | -2.5473 | -1.9641 |
|  | (-0.0315) | (-0.3576) |
| Industry-year fixed effects | Yes | Yes |
| N | 27930 | 27930 |
| R-sq | 0.7653 | 0.6954 |
| adj. R-sq | 0.7754 | 0.6642 |

variable (*func_rich*), and it is independent of other variables that cannot be observed but will affect the dependent variable (firm performance, *ROA*). This paper uses the average CEOs' functional richness (*func_mean*) in the same province and industry in the same year as an instrumental variable of the *func_rich*.

Table 5 reports the regression results using the instrumental variable. Column (1) of Table 5 reports the regression results of the first stage of instrumental variables. It can be seen from column (1) of Table 5 that the *func_mean* regression coefficient is significantly positive at 1%, meeting the correlation conditions. At the same time, Cragg Donald Wald F statistic result shows 1736.93, and Kleibergen Paap Wald rk F statistic result shows 1006.22, both of which reject the hypothesis of weak instrumental variables. Column (2) of Table 5 reports the regression results of the second stage of instrumental variables. It can be seen from column (2) of Table 5 that the *func_rich* regression coefficient is significantly positive at the level of 1%,

which indicates that the research conclusion of this paper is still valid after using instrumental variables to solve potential endogenous problems.

## 6. Conclusion and insights

Based on the upper echelons theory and the career focus theory, this paper collected and compiled unique data sets related to the CEOs' functional experience of A-share listed companies in Shanghai and Shenzhen, China, from 2006 to 2020. The artificial intelligence big data method was used to construct CEOs' breadth and depth indicators. The relationship between CEOs' functional experience and firm performance was theoretically explained, analyzed, and validated. The study results show that as a CEO's functional experience richness increases, there is a u-shaped change in the relationship between it and firm performance that first decreases and then increases. As the depth of a CEO's tenure increases from weak to strong, the relationship between their tenure and firm performance increases and decreases in an inverted u-shape. The findings of this paper can provide empirical evidence to support the study of firm performance.

A CEO's functional experience is increasingly mentioned in studies of firm performance. However, there is still a lack of theoretical arguments about how tenure characteristics affect firm performance. This paper aimed to provide a more specific reference for the research on the impact of CEOs' functional experience on firm performance to bridge the existing research gap. The possible theoretical contributions of this paper are as follows.

First, introducing the breadth and depth characteristics of CEOs' functional experience further enriches the study of the upper echelons theory. The theory emphasizes the important impact of functional experience on firm performance. However, it does not construct functional experience characteristics at a deeper level. An increasing number of contemporary practices equate job category and tenure with functional experience, ignoring the important role of tenure and rank for each position segment. This leads to a rich or exclusive functional experience impact in the empirical test. Therefore, this paper introduced and empirically tested the characteristics of both the breadth and depth of CEOs' functional experience and proposed that these two dimensions influence firm performance. Second, in addition to the direct effect of the two dimensions of experience on firm performance, this paper further emphasized the nonlinear relationship between the two dimensions. It expands the literature on the relationship between CEOs' experience and firm performance.

This paper also has important managerial implications. For shareholders looking for long-term capital returns, this paper explains what type of CEOs are better suited for public companies in China. In addition, understanding CEOs' functional experience helps managers to make corporate management decisions. With the deepening of global diversification, the battle for talent has become more intense. This paper provides a theoretical basis for constructing a hierarchy of talented teams based on the depth of tenure. The depth of tenure of different CEOs at the senior, intermediate, and junior levels varies. Only when the functional experience of the CEO matches the needs of the company can the quality of talent be better enhanced and the company's performance improved. In addition, based on the Chinese context, this paper used Chinese-listed companies as the research sample. The importance of CEOs' experience was discussed in detail. It not only provides evidence for the selection of high-level talents but also helps to further standardize a company's employee structure system. In turn, it promotes the sustainable development of the whole society.

This paper also has limitations. First, a CEO's functional experience is an important characteristic of firm performance. However, considering the many factors influencing firm performance, this paper covered the control variables as much as possible, but there still may be

missing factors. Second, this paper used CEOs' resume data to extract tenure information from text. However, the text in a resume cannot fully represent the real functional experience of individuals, leading to possible errors in the portrayal of functional experience in this paper. Future research may improve this paper's measurement of CEOs' functional experience through multi-dimensional and more fine-grained features.

## Author Contributions

**Conceptualization:** Xiaohong Huang, Xiaorong Zhang.

**Data curation:** Xiaohong Huang, Jiangwei Liu.

**Investigation:** Xiaohong Huang.

**Methodology:** Xiaohong Huang, Jiangwei Liu, Qianqian Zeng.

**Resources:** Liangyu Min, Xiaorong Zhang.

**Software:** Xiaohong Huang, Jiangwei Liu, Liangyu Min, Qianqian Zeng.

**Supervision:** Liangyu Min, Jun Zhang.

**Validation:** Qianqian Zeng.

**Visualization:** Xiaohong Huang, Jun Zhang.

**Writing – original draft:** Xiaohong Huang.

**Writing – review & editing:** Xiaohong Huang, Jun Zhang, Xiaorong Zhang.

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
