## [Decision Letter · Decision Letter 0]

22 Jul 2022

PONE-D-22-03262CEO's Functional Experience and Firm Performance Based on Text MiningPLOS ONE

Dear Dr. Huang,

Thank you for submitting your manuscript to PLOS ONE. After careful consideration, we feel that it has merit but does not fully meet PLOS ONE’s publication criteria as it currently stands. Therefore, we invite you to submit a revised version of the manuscript that addresses the points raised during the review process.

ACADEMIC EDITOR:The reviewers are experts in the area and have prepared a careful and fair review. I do appreciate your efforts in writing the manuscript and I find the topic interesting and worth pursuing. However, as you can see from the report, the reviewers have different evaluations (and recommendation) on the quality of your paper (one revision and one rejection). I therefore have considered it myself taking into account the comments from reviewers and decided to give you an opportunity to revise your manuscript as I believe you can address them properly. I hope you will take this chance to improve the quality of the paper to satisfy both reviewers. Specifically, I would suggest you to carefully follow the comments to revise your manuscript and resubmit it for re-consideration for publication. To be published, you should address at least the following summary of key points as below. However, please provide full responses to all comments of two reviewers. Please find detailed comments in the review reports.

1. Adding a new hypothesis that is exactly the same (H1, H2) which deals with the female CEO breath and depth, you made a huge effort to sort the data over 25000 observations covered the period 2006-2020 which is good. As the reviewer suggested, if you faced serious difficulty to grab/splitting the same data into a female context then you may continue as it is. But I would suggest you consider this point as it may be adding good value to your manuscript.

2. Add discussions about the diagnostics test

3. The introduction section of the paper need to clearly address the motivation of the paper.

4. Extend the literature review part by including more studies which directly investigate the role of CEO experience in determining firm performance.

5. Regarding Section 3.1, you claim that CEO’s functional experience increases the “instability” and this instability negatively affects the firm performance. The reviewer is not quite confident about this argument. What does “instability” refer to? Is it instability in deciding corporate policies or in financial performance? Also it is not clear why CEO experience increases instability.

6. Related to the previous comment, you also state that “This instability affects the company’s performance and has a greater risk-averse ability”. This argument is also not clear. The reviewer does not quite sure about what “greater risk-averse ability” means? They think you should make your arguments more clear and direct when setting up the hypotheses. It is also essential to support these arguments with the literature.

7. It is useful to consider Tobin’s Q as the market-based performance measure in addition to the ROA and ROE which are accounting based measures.

8. The reviewer does not comfortable with the data definitions. You use (total profit + interest expense) scaled by Average total assets. Not sure why you add interest expense to the total profit. Also, ROE definition is not correct. You scale net profit by net assets to calculate ROE. It should be scaled by Equity. The reviewer suggests you to follow the papers published in reputable journals when defining the variables.

9. Explain how PSM has been employed. Considering the number of observations for each Column in Table 3, you manage to match each firm in the treated group with another firm in the control sample because there is no decrease in the number of observations which is quite uncommon in PSM analysis. Then, what is the difference between Column 3 and Column 5 in Table 3?

10.Section 5.2.2 is also confusing. Is there any definition for KMO? You state that first principal component explains 76.48% of the total variables. How did you end up with this number? Are PEF (in the text) and PET (in Table 3) the same? You should revise this part to avoid any confusion.

11. Check reference list for any mistakes, and address other minor comments from reviewers such as citations, typos and grammatical errors.

We look forward to receiving your revised manuscript.

Kind regards,

Vu Quang Trinh, PhD

Academic Editor

PLOS ONE

2. PLOS ONE does not copy edit accepted manuscripts (https://journals.plos.org/plosone/s/criteria-for-publication#loc-5). To that effect, please ensure that your submission is free of typos and grammatical errors.

Reviewers' comments:

Reviewer's Responses to Questions

**Comments to the Author**

1. Is the manuscript technically sound, and do the data support the conclusions?

Reviewer #1: Partly

Reviewer #2: Yes

2. Has the statistical analysis been performed appropriately and rigorously? 

Reviewer #1: No

Reviewer #2: Yes

3. Have the authors made all data underlying the findings in their manuscript fully available?

Reviewer #1: Yes

Reviewer #2: Yes

4. Is the manuscript presented in an intelligible fashion and written in standard English?

Reviewer #1: No

Reviewer #2: Yes

5. Review Comments to the Author

Reviewer #1: In this paper, authors investigate the impact of CEO’s functional experience on firm performance using a text mining approach. They find that CEO’s breadth and depth in terms of experience and background have nonlinear impacts on firm performance. I found the paper interesting but have several major concerns particularly about the motivation and methodology which were stated below.

1) The introduction section of the paper does not clearly address the motivation and the contribution of the paper. As authors state in the introduction, there are several studies in the literature which investigate the role of CEO background or experience on firm performance or other financial policies. However, it is quite not clear in the introduction that in which ways this paper differentiates from the other papers in the literature. Authors state that “the purpose of the paper was to increase the interest of researchers in continuing to study CEO’s functional experience characteristics and their effects on firm performance”. I am not quite sure if this statement clearly identifies the objectives of research. What are the specific aims or research questions of the study and how they contribute to the literature? Authors also claim that the main contribution of the study is using a “cutting-edge information technology methods”. I think authors should also extend this part by a few sentences to make the contribution of the paper clearer in the introduction. Overall, authors could revise the introduction section to make it more informative for the readers.

2) Authors can extend the literature review part by including more studies which directly investigate the role of CEO experience in determining firm performance. For example, Hamori and Koyuncu (2015) investigate the impact of prior CEO experience on firm performance. In addition, Li and Patel (2019) investigate the impact of CEO generalist experience and firm performance. Also, Georgakakis and Ruigrok (2017) provide evidence on the relationship CEO characteristics and background, and organizational performance. I couldn’t find any discussion about these studies. It would be useful to extend the literature by including more studies.

References

Hamori, M., & Koyuncu, B. (2015). Experience matters? The impact of prior CEO experience on firm performance. Human Resource Management, 54(1), 23-44.

Li, M., & Patel, P. C. (2019). Jack of all, master of all? CEO generalist experience and firm performance. The Leadership Quarterly, 30(3), 320-334.

Georgakakis, D., & Ruigrok, W. (2017). CEO succession origin and firm performance: A multilevel study. Journal of Management Studies, 54(1), 58-87.

3) Regarding Section 3.1, authors claim that CEO’s functional experience increases the “instability” and this instability negatively affects the firm performance. I am not quite confident about this argument. What does “instability” refer to? Is it instability in deciding corporate policies or in financial performance? Also it is not clear why CEO experience increases instability.

4) Related to my previous comment, authors also state that “This instability affects the company’s performance and has a greater risk-averse ability”. This argument is also not clear. I am not quite sure about what “greater risk-averse ability” means? I think authors make their arguments more clear and direct when setting up the hypotheses. It is also essential to support these arguments with the literature.

5) Authors use ROA and ROE as the performance measures. It is useful to consider Tobin’s Q as the market-based performance measure in addition to the ROA and ROE which are accounting based measures.

6) I am not comfortable with the data definitions. Authors use (total profit + interest expense) scaled by Average total assets. Not sure why authors add interest expense to the total profit. Also, ROE definition is not correct. Authors scale net profit by net assets to calculate ROE. It should be scaled by Equity. I suggest authors to follow the papers published in reputable journals when defining the variables.

7) Authors employ PSM to address selection bias and the results are reported in Table 3 and Column 5. However, it is not clear how PSM has been employed. Considering the number of observations for each Column in Table 3, authors manage to match each firm in the treated group with another firm in the control sample because there is no decrease in the number of observations which is quite uncommon in PSM analysis. Then, what is the difference between Column 3 and Column 5 in Table 3?

8) Section 5.2.2 is also confusing. Is there any definition for KMO? Authors state that first principal component explains 76.48% of the total variables. How did authors end up with this number? Are PEF (in the text) and PET (in Table 3) the same? Authors should revise this part to avoid any confusion.

9) Authors should check the reference list for any mistakes. For example, Reference 48 has been published in “Journal of Finance and Bank Management” not in the “Journal of Finance”. It would be useful to check references to avoid any typos and mistakes in the Journal names.

10) I also recommend authors to avoid citing papers in unknown journals.

11) There are several typos in the text. It would be useful to review/proofread the text to avoid any typos and grammatical errors.

Reviewer #2: I am happy to review this research work. It is unique in nature and offers novel findings, particularly in the Chinese context. Overall it's well written, while I have only one suggestion which makes further interesting finds of this work, that is the female CEO. The author(s) explained well the variable measurement along with gender classification (male 1, female 0). I suggest adding a new hypothesis that is exactly the same (H1, H2) which deals with the female CEO breath and depth, Author(s) made a huge effort to sort the data over 25000 observations covered the period 2006-2020 which is good. This is just my suggestion, if the author(s) faced serious difficulty to grab/splitting the same data into a female context then the author(s) may continue as it is.

Additionally, Indeed, it is new work, with novel findings while the author(s) may add some recent literature to further strengthen the study. Currently, there is no study referring to 2022 and very few studies refer to 2021.

The author (s) used an industry-year fixed effect in the model which seems that the author used a fixed effect model rather than random or others. There is no discussion about the diagnostics test, I believe it must be checked during the model/analysis selection but if it is mentioned in the study it also strengthens the analysis/results justifications.

The study is well concluded along with insights and limitations while the future research question may increase which opens new research steam for other scholars.

Overall, the author(s) did a good job, I recommend this work for publication in PLOS ONE subject to the suggested minor additions.

6. PLOS authors have the option to publish the peer review history of their article (what does this mean?). If published, this will include your full peer review and any attached files.

Reviewer #1: No

Reviewer #2: No

---

## [Author Response · Author response to Decision Letter 0]

4 Sep 2022

Dear Editor,

Thank you for your valuable comments and suggestions on the structure of our manuscript. We have modified the manuscript accordingly, and the main corrections in the paper and the responses to the referees' comments are as follows:

Reviewer #1

Q1: The introduction section of the paper does not clearly address the motivation and the contribution of the paper. As authors state in the introduction, there are several studies in the literature which investigate the role of CEO background or experience on firm performance or other financial policies. However, it is quite not clear in the introduction that in which ways this paper differentiates from the other papers in the literature. Authors state that “the purpose of the paper was to increase the interest of researchers in continuing to study CEO’s functional experience characteristics and their effects on firm performance”. I am not quite sure if this statement clearly identifies the objectives of research. What are the specific aims or research questions of the study and how they contribute to the literature? Authors also claim that the main contribution of the study is using a “cutting-edge information technology methods”. I think authors should also extend this part by a few sentences to make the contribution of the paper clearer in the introduction. Overall, authors could revise the introduction section to make it more informative for the readers.

Re: We apologize for the poorly considered experiment. Accordingly, we added the motivation of the paper. 

The relationship between CEO function experience and firm performance seems to be a black box problem. Thus, how to further analyzing this problem has motivated the researchers to carry out this study. Many studies support the idea that CEO experience affects firm performance. However, there are still some shortcomings in the existing research. (1) The breadth-based research does not consider the diversity and duplication of functions, and the rough count of the number of positions is not enough to reveal the breadth of the CEO's service experience. (2) The depth-based research does not consider the tenure of each stage of employment, and these studies do not have a hierarchy of positions. (3) Current data analysis and processing methods are not always scalable, especially manual processing methods cannot automate research on a wide range of listed companies. Hence, these factors motivated the researchers to conduct this study.

Q2: Authors can extend the literature review part by including more studies which directly investigate the role of CEO experience in determining firm performance. For example, Hamori and Koyuncu (2015) investigate the impact of prior CEO experience on firm performance. In addition, Li and Patel (2019) investigate the impact of CEO generalist experience and firm performance. Also, Georgakakis and Ruigrok (2017) provide evidence on the relationship CEO characteristics and background, and organizational performance. I couldn’t find any discussion about these studies. It would be useful to extend the literature by including more studies..

Re: Thank you very much for your suggestion. We have carefully read the literature you provided and added it to our paper. At the same time, we add some recent literature to strengthen the study further.

Q 3: Regarding Section 3.1, authors claim that CEO’s functional experience increases the “instability” and this instability negatively affects the firm performance. I am not quite confident about this argument. What does “instability” refer to? Is it instability in deciding corporate policies or in financial performance? Also it is not clear why CEO experience increases instability.

Re: We are mainly referring to the instability of the environment and decision-making. Recent studies have found that CEO's experience hinders subsequent firm performance (44, 45). The rationale is that prior experience is too heavily laden with the specific environments in which it was gained and therefore is not as beneficial to the new firms as the CEOs believe it will be. In short, available evidence suggests that past experience is a detriment to success in a subsequent job, mainly because experienced CEOs have to give up firm-specific skills that are not useful to a new firm (45). It is exacerbated by the tendency of CEOs to become less adaptable with increasing experience (40, 46, 47). Therefore, as the CEO's functional experience increases, they face more and more instability in the environment and decision-making, which is not conducive to improving the firm performance (48).

Q4: Related to my previous comment, authors also state that “This instability affects the company’s performance and has a greater risk-averse ability”. This argument is also not clear. I am not quite sure about what “greater risk-averse ability” means? I think authors make their arguments more clear and direct when setting up the hypotheses. It is also essential to support these arguments with the literature.

Re: The upper echelons theory believes that the individual's experience characteristics also affect risk preference (1). When a listed company hires a CEO, their experience is usually considered. In reality, more and more listed companies hire experienced CEOs because they can provide more professional policy advice for the company's development. On the one hand, experienced CEOs can apply their expertise to practical decisions, helping to make sound decisions. On the other hand, reliable and prudent practice requirements will also prompt them to form corresponding characters and habits with a lower risk appetite (49).

Q5: Authors use ROA and ROE as the performance measures. It is useful to consider Tobin’s Q as the market-based performance measure in addition to the ROA and ROE which are accounting based measures.

Re: Tobin's Q: Ratio of the market value of the firm to total assets. The regression results of the Tobinq indicator are added to the robustness test of this paper. The specific results are shown in Table 3.

Q6: I am not comfortable with the data definitions. Authors use (total profit + interest expense) scaled by Average total assets. Not sure why authors add interest expense to the total profit. Also, ROE definition is not correct. Authors scale net profit by net assets to calculate ROE. It should be scaled by Equity. I suggest authors to follow the papers published in reputable journals when defining the variables.

Re: Thank you for your carefully and patiently reviewing the manuscript. We are sincerely sorry for our manuscript's incorrect definition of those variables. We have already corrected the description according to your valuable suggestion in the revised manuscript. 

ROA: Return on Assets i.e. the ratio of total income and total assets.

ROE: Return on Equity i.e. ratio of net income to shareholder's equity.

Q7: Authors employ PSM to address selection bias and the results are reported in Table 3 and Column 5. However, it is not clear how PSM has been employed. Considering the number of observations for each Column in Table 3, authors manage to match each firm in the treated group with another firm in the control sample because there is no decrease in the number of observations which is quite uncommon in PSM analysis. Then, what is the difference between Column 3 and Column 5 in Table 3?

Re: Our paper uses the sample median of func_rich as the benchmark to divide the sample into experimental and control groups for the PSM test. After careful consideration, it is found that there is indeed a problem with the PSM made in this article. To further address this issue, we propose a robustness check that we believe is better.

A potential problem with our findings is that listed companies have other executives besides the CEO. According to our statistics, the top management team changes almost every year. If the top management team is highly fluid, it is difficult to measure whether the CEO or other executives bring about an improvement in firm performance.

Therefore, this paper analysis the stability of the CEO and top management team each year. Figure 1 shows the CEO and top management team changes over the statistical period as a group of company representatives. The blue dotted line indicates whether the CEO has changed in the current year, 1 if the CEO changes, and 0 if there is no change. The solid red line represents the rate of change in the top management team.

Figure 1 The change rate of the CEO and Top Management Team

 [image/table included in attached document]

As can be seen from the figure, most listed companies have changed their CEOs during the statistical period. The changes in the top management team are different. Some companies have undergone substantial changes in the top management team during the statistical period, while the changes were relatively small in others. We further makes a descriptive statistic of the change rate of the top management team. The rate of change of the top management team is distributed between [0, 1]. The average rate was 13%. Therefore, this paper believes that if the current year's change rate of the top management team is less than 13%, it is relatively stable. According to this standard, this paper filtered the data of CEO changes and a relatively stable top management team for model testing. The part of sample are shown in Figure 2.

Figure 2 The change rate of the CEO and Top Management Team in the filtered samples

 [image/table included in attached document]

The filtered samples are smaller but cleanest. In these samples, it is guaranteed that the CEO changes, but the top management team remains relatively stable. This method can better describe the relationship between CEOs' functional experience and company performance. The model empirical analysis results for this sample are in column 7 in Table 3. The model results still support Hypotheses 1 and 2.

Q8: Section 5.2.2 is also confusing. Is there any definition for KMO? Authors state that first principal component explains 76.48% of the total variables. How did authors end up with this number? Are PEF (in the text) and PET (in Table 3) the same? Authors should revise this part to avoid any confusion.

Re: PEF (in the text) and PET (in Table 3) the same. We have unified. The KMO (Kaiser-Meyer-Olkin) test statistic is an indicator used to compare simple and partial correlation coefficients between variables. It is mainly used in factor analysis of multivariate statistics. We are very sorry for confusing the two concepts, this paper used PCA (Principal Component Analysis) to analyze the indicators.

PCA is to try to recombine the original many related indicators (such as P indicators) into a new set of comprehensive indicators independent of each other to replace the original ones. The results of the principal component analysis are shown in Table 4. Among them, (Component) represents the name of the principal component extracted by the system. It can be found that a total of 5 principal components have been extracted. The Eigenvalue column represents the eigenvalue of the principal component extracted by the system. The size of the eigenvalue means the explanatory power of the principal component. The larger the eigenvalue, the stronger the explanatory power. The Proportion column represents the variance contribution rate of the principal component extracted by the system, and the variance contribution rate is also the explanatory power of the principal component.

Table 4 Regression results

 [image/table included in attached document]

Ultimately, the characteristic root of the first principal component is 2.4820, which is greater than 1. The first principal component explains 82.73% of the total variables. The expression of the composite indicator of firm performance (PEF) obtained by expressing the first principal component as a linear combination of the individual variables is:

PEF=0.5861ROA+0.6049ROE+0.5392TobinQ (5)

The empirical test was conducted with TobinQ and PEF as the dependent variable. As shown in column (5) and (6) in Table 3, the regression results still support hypothesis one and hypothesis two.

Q9: Authors should check the reference list for any mistakes. For example, Reference 48 has been published in “Journal of Finance and Bank Management” not in the “Journal of Finance”. It would be useful to check references to avoid any typos and mistakes in the Journal names.

Re: In the last version, Reference 48 is: Azeez A A. Corporate governance and firm performance: evidence from Sri Lanka[J]. Journal of Finance, 2015, 3(1): 180-189. The reference is indeed from “Journal of Finance”. To avoid ambiguity, we removed this reference.

Q10: I also recommend authors to avoid citing papers in unknown journals.

Re: Thanks for your suggestion, we have taken your advice and updated the references.

Q11: There are several typos in the text. It would be useful to review/proofread the text to avoid any typos and grammatical errors.

Re: We have a try to correct the spelling errors and paragraphs to make them more readable.

Reviewer #2:

Q1: I am happy to review this research work. It is unique in nature and offers novel findings, particularly in the Chinese context. Overall it's well written, while I have only one suggestion which makes further interesting finds of this work, that is the female CEO. The author(s) explained well the variable measurement along with gender classification (male 1, female 0). I suggest adding a new hypothesis that is exactly the same (H1, H2) which deals with the female CEO breath and depth, Author(s) made a huge effort to sort the data over 25000 observations covered the period 2006-2020 which is good. This is just my suggestion, if the author(s) faced serious difficulty to grab/splitting the same data into a female context then the author(s) may continue as it is. 

Re: We do appreciate your comments. With the rapid development of the social economy, more and more female managers have entered top management and achieved excellent results. From the perspective of female CEOs, studying their impact on firm performance is not only a realistic response to contemporary corporate research but also reflects the demand trend of contemporary research. With the improvement of female status, especially the rapid increase in female CEOs in modern firms, it is of great theoretical and practical significance to study the impact of female CEOs' functional experiences on firm performance.

In China, however, females often work harder to achieve the same status as males. Females' ability to work is often underestimated, and not many of them can become CEOs of public companies. Through the statistical analysis of the data in this paper, in the 15-year statistical period, female CEOs only accounted for 6.4%, and the ratio of males and females was seriously unbalanced.

We split the data into male and female CEOs, then analyzed the breadth and depth of female and male CEOs' functional experiences and found some interesting results. First, Figures 3 shows the descriptive statistics of the two subsamples. Figures 4 and 5 are the position promotion charts for female and male CEOs. We found those female CEOs had a wider breadth of functional experience than males, with an average of 0.9054 but 0.5159 for males. However, in terms of depth of experience (level_score and avg_emp), female CEOs generally have a lower depth of experience.

Figure 3 Descriptive statistics of the two subsamples

 [image/table included in attached document]

Figure 4 Evolution of CEO's position promotion chart of female CEOs

 [image/table included in attached document]

Figure 5 Evolution of CEO's position promotion chart of male CEOs

 [image/table included in attached document]

The above is only a preliminary exploration, the comments are of great value to this paper, and we also found some highlights of the exploration. However, we believe that analyzing female and male CEOs still faces many problems and needs complete data and theoretical support. Thanks for your suggestion; this is what we need to work on in the future. We are also working on this; you will see research in this area in our future work.

Q2: Indeed, it is new work, with novel findings while the author(s) may add some recent literature to further strengthen the study. Currently, there is no study referring to 2022 and very few studies refer to 2021.

Re: Thank you very much for your suggestion. We have added some recent literature to strengthen our study.

Q 3: The author (s) used an industry-year fixed effect in the model which seems that the author used a fixed effect model rather than random or others. There is no discussion about the diagnostics test, I believe it must be checked during the model/analysis selection but if it is mentioned in the study it also strengthens the analysis/results justifications.

Re: This is a constructive suggestion to improve the paper's publication potential. According to the characteristics of panel data, we test the validity of the fixed-effects model and the random-effects model on the validity of the regression model. Usually, the Hausman test is used to judge the applicability of the two models (see Figure 6). The p-value of the Hausman test result is 0.0000, so the null hypothesis is strongly rejected, and a fixed-effects model should be used instead of a random-effects model.

Figure 6 The result of Hausman test

 [image/table included in attached document]

Thank you for your patiently reviewing the manuscript. We are sincerely sorry for our carelessness. We have already corrected the description according to your valuable suggestion in the revised manuscript. We hope that you will be satisfied with the revision. Subsequently, we look forward that the revised manuscript being adaptable for the demands for open publication.

I want to re-submit this manuscript and hope it is acceptable for publication in the journal. If there are any questions about our paper, please do not hesitate to let us know. Thank you very much for your attention to our paper.

Best regards,

Xiaohong Huang

School of Logistics and Management Engineering

Yunnan University of Finance and Economics

Kunming, 650221, China

E-mail: huangxiaohong@163.sufe.edu.cn

---

## [Decision Letter · Decision Letter 1]

12 Oct 2022

PONE-D-22-03262R1CEO's Functional Experience and Firm Performance Based on Text MiningPLOS ONE

Dear Dr. Huang,

Thank you for submitting your manuscript to PLOS ONE. After careful consideration, we feel that it has merit but does not fully meet PLOS ONE’s publication criteria as it currently stands. Therefore, we invite you to submit a revised version of the manuscript that addresses the points raised during the review process.

ACADEMIC EDITOR: Thank you for your efforts in revising your manuscript following two reviewers' comments. As you can see from the report, one reviewer has recommended an acceptance for publication while another still has some constructive suggestions for improvement. I therefore decided to give you an opportunity to revise your manuscript (minor revision) as I believe you can address them properly (as best as you can). I hope you will take this chance to improve the quality of the paper to satisfy the reviewer. Please note that you don't have to worry about the first comment related to the contribution but if you can, try to clarify the motivation (I encourage you doing so) and a bit of contribution for future readers (but not compulsory). Please provide full responses to comments of the reviewer. I will not repeat them here to avoid your confusions. Please find detailed comments in the review reports. To avoid further delay, at this stage, I would encourage you using professional proof reading service to improve the writing style and correct grammatical and spelling errors. 

We look forward to receiving your revised manuscript.

Kind regards,

Vu Quang Trinh, PhD

Academic Editor

PLOS ONE

Journal Requirements:

Additional Editor Comments (if provided):

NA

Reviewers' comments:

Reviewer's Responses to Questions

**Comments to the Author**

1. If the authors have adequately addressed your comments raised in a previous round of review and you feel that this manuscript is now acceptable for publication, you may indicate that here to bypass the “Comments to the Author” section, enter your conflict of interest statement in the “Confidential to Editor” section, and submit your "Accept" recommendation.

Reviewer #1: (No Response)

Reviewer #2: All comments have been addressed

2. Is the manuscript technically sound, and do the data support the conclusions?

Reviewer #1: Partly

Reviewer #2: Yes

3. Has the statistical analysis been performed appropriately and rigorously? 

Reviewer #1: No

Reviewer #2: Yes

4. Have the authors made all data underlying the findings in their manuscript fully available?

Reviewer #1: Yes

Reviewer #2: Yes

5. Is the manuscript presented in an intelligible fashion and written in standard English?

Reviewer #1: No

Reviewer #2: Yes

6. Review Comments to the Author

Reviewer #1: Authors made some improvements in the paper based on the comments provided in the previous round. However, I still have some concerns regarding particularly data and methodology section which were stated below.

1) Introduction section now clearly states the motivation and the contributions of the paper.

2) It would be useful to add references for calculating func_rich and level_score. For example, authors use a weighted score of 6:3:1 for senior, middle, and junior levels. Is there any reason for using a weight of 6 for senior levels and 3 for middle levels? As they are the main variables of interest, authors should be more precise and clearer when explaining these variables.

3) ROA explanation is still not correct. It should be “the ratio of net income and total assets” not “total income”.

4) Degree variable is a categorical variable, but it is included in the regression as a continuous variable (as I understand from the results table). Authors should check which variables are continuous and categorical.

5) It is good to include Tobin’s Q variable as another proxy for the firm performance. However, there is no discussion about it. How is it calculated? Is it only an alternative to ROA and ROE (for robustness purposes) or another performance proxy such as PEF? It would be useful to provide more discussion/explanations about it.

6) Authors claim that they are using firm fixed effects regression in Section 4.3 based on Hausman test, but Table 3 suggests “Industry fixed effects”. I am not quite sure if authors use firm or industry fixed effects.

7) I am still not convinced about the endogeneity test. Authors provide some graphs (descriptive) about the relationship between CEO and top management teams changes, but it is not clear how it addresses endogeneity. What should readers understand from these graphs? Are there any other sources of endogeneity such as omitted variables, selection bias etc.? After explaining the Figure 2, authors state that “the model results still support Hypothesis 1 and 2”. I am not quite sure which model authors refer to and how it supports Hypothesis 1 and 2.

Reviewer #2: The author (s) carefully incorporate all the suggestion. I am happy to recommend it for publication in PLOS ONE.

7. PLOS authors have the option to publish the peer review history of their article (what does this mean?). If published, this will include your full peer review and any attached files.

Reviewer #1: No

Reviewer #2: No

---

## [Author Response · Author response to Decision Letter 1]

9 Dec 2022

Dear Editor,

Thank you for your valuable comments and suggestions on the structure of our manuscript. We have modified the manuscript accordingly. 

Q1: Please note that you don't have to worry about the first comment related to the contribution but if you can, try to clarify the motivation (I encourage you doing so) and a bit of contribution for future readers (but not compulsory). 

Re: Thank you very much for your suggestion. We have carefully read some relation literature and clarify the motivation and contribution of our paper as follow.

Motivation : From a simple statistical view, some representative companies, such as Tencent and Alibaba, have achieved excellent firm performance, but the experience of their CEOs is very different. Moreover, there are also other CEOs whose experience falls somewhere in between. From the CEOs' view, we were interested in the differences in their experience. At the same time, it is also worth paying attention to the direction of the firm performance results of these differences. The question arises: What characteristics of a CEO's experience are more conducive to improved firm performance? Examining this question may enable us to better understand the motivation for CEO selection and facilitate a reasonable and targeted CEO evaluation.

Contributions: Our study makes three contributions to the literature. First, compared with the subjective judgment of manual collation and questionnaires, this paper processed the data using cutting-edge information technology methods such as named entity recognition. We collected CEOs' resume data for 15 consecutive years from 2006 to 2020 to obtain comprehensive experience details. The analysis results are scientific and objective, and the technical approach can be extended to other types of executives. Second, compared with studies that mainly focus on the job category and tenure of executives' experience, this article introduced the breadth and depth dimensions of CEOs' functional experience enriching the study of the upper echelons theory. Finally, a few studies have provided partial evidence of the relationship between a CEO's experience and firm performance. In addition to the direct effect of the CEO's experience on firm performance, this paper further emphasized the nonlinear relationship between the two dimensions. The findings of this paper may serve as direct evidence for related studies to expand the literature on CEOs' experiences.

Q2: To avoid further delay, at this stage, I would encourage you using professional proof reading service to improve the writing style and correct grammatical and spelling errors. 

Re: Thank you for your patiently reviewing the manuscript. We are sincerely sorry for our carelessness. We have already corrected the description according to your valuable suggestion in the revised manuscript.

Reviewer

The main corrections in the paper and the responses to the reviewer's comments are as follows:

Q1: Introduction section now clearly states the motivation and the contributions of the paper.

Re: Thank you very much for your affirmation. At the same time, combined with the comments of the editor, we further clarified the motivation and the contributions.

Q2: It would be useful to add references for calculating func_rich and level_score. For example, authors use a weighted score of 6:3:1 for senior, middle, and junior levels. Is there any reason for using a weight of 6 for senior levels and 3 for middle levels? As they are the main variables of interest, authors should be more precise and clearer when explaining these variables .

Re: We apologize for the poorly considered explanation. Accordingly, we added the explanation of the weighted score. On August 31, 2006, the Ministry of Human Resources and Social Security of the People's Republic of China issued the Implementation Opinions of the Tentative Measures for the Management of Post Establishment in Public Institutions. According to Article 13 of the setting of professional and technical post levels states: the structural proportion of senior, intermediate, and junior posts shall be controlled according to the regional economic and social development level and industrial characteristics, as well as the function, specification, subordination and professional and technical level of the public institution. The overall national control target is 1:3:6 for the proportion of senior, intermediate, and junior posts. 

Combined with the structural ratio of senior, intermediate, and junior positions, this paper gave a weighted score of 6:3:1 to senior, intermediate, and junior levels.

We have added references and links as follows:

[56]MoP. Administrative Measures for Post Setting in Public Institutions. Administrative Measures for Post Setting in Public Institutions. 2006(70). http://www.mohrss.gov.cn/SYrlzyhshbzb/rencairenshi/zcwj/shiyedanweirenshiguanli_1/202002/t20200210_358464.html

Q3: ROA explanation is still not correct. It should be “the ratio of net income and total assets” not “total income” .

Re: Thank you for your patiently reviewing. We are sincerely sorry for our carelessness. We have already corrected the explanation of ROA according to your valuable suggestion in the revised manuscript.

Q4: Degree variable is a categorical variable, but it is included in the regression as a continuous variable (as I understand from the results table). Authors should check which variables are continuous and categorical .

Re: In this paper, we control the CEO's educational background in the regression model. Higher education can improve CEOs' overall quality and professionalism, making them more decisive and proactive in enterprise management. In the opinion of the Board of Directors, such CEOs tend to be more capable, dare to take risks, and make correct decisions, which may be conducive to improving firm performance. In addition, when choosing CEOs, enterprises also trust CEOs with a better educational background more and give them more rights. Therefore, CEOs with better educational background have more say in improving firm performance. The selection of educational background in this paper is the highest degree of CEO. The highest education refers to the highest degree (such as junior college degree, bachelor's degree, master's degree, or doctor's degree) that the CEO has obtained up to the time the sample is used. The higher the degree, the higher the degree score. According to the research of Fan et al., CEO education level is a score ranging between 1 and 5: 1 denotes a degree below junior college; 2 denotes a junior college degree; 3 denotes a bachelor's degree; 4 denotes a master's degree; 5 denotes a doctoral degree.

Fan JP, Wong TJ, Zhang T. Politically connected CEOs, corporate governance, and Post-IPO performance of China's newly partially privatized firms. J Financ Econ. 2007;84(2):330-57.

Q5: It is good to include Tobin’s Q variable as another proxy for the firm performance. However, there is no discussion about it. How is it calculated? Is it only an alternative to ROA and ROE (for robustness purposes) or another performance proxy such as PEF? It would be useful to provide more discussion/explanations about it.

Re: In the robustness test, we use Tobin's q measure of firm performance in the robustness test as our dependent variable. Tobin's q ratio (or q as it is commonly known), defined as the capital market value of the firm divided by the replacement value of its assets incorporates a market measure of firm value which is forward-looking, risk-adjusted, and less susceptible to changes in accounting practices (64, 65).

[64] Bharadwaj AS, Bharadwaj SG, Konsynski BR. Information technology effects on firm performance as measured by Tobin's q. Manage Sci. 1999;45(7):1008-24.

[65] Montgomery CA, Wernerfelt B. Diversification, Ricardian rents, and Tobin's q. The Rand journal of economics. 1988:623-32.

Q6: Authors claim that they are using firm fixed effects regression in Section 4.3 based on Hausman test, but Table 3 suggests “Industry fixed effects”. I am not quite sure if authors use firm or industry fixed effects.

Re: This paper adopts the two-way fixed effects model. Yeardummy and Industrydummy represent the year dummy and industry dummy variables, respectively. Both of them are 0-1 dummy variables. uit is the model error term. i is the i - th firm. t is the t year. The above control and dummy variables are added to the model to control the influence of other factors further.

Q7: I am still not convinced about the endogeneity test. Authors provide some graphs (descriptive) about the relationship between CEO and top management teams changes, but it is not clear how it addresses endogeneity. What should readers understand from these graphs? Are there any other sources of endogeneity such as omitted variables, selection bias etc.? After explaining the Figure 2, authors state that “the model results still support Hypothesis 1 and 2”. I am not quite sure which model authors refer to and how it supports Hypothesis 1 and 2 .

Re: First, explanation of the graphs is as follows: A potential endogeneity problem with our findings is that listed companies have other executives besides the CEO. According to our statistics, the top management team changes almost every year. If the top management team is highly fluid, it is difficult to measure whether the CEO or other executives bring about an improvement in firm performance.

Therefore, we overcame the obstacles and selected the cleanest sample that was as undisturbed as possible. In these samples, it is guaranteed that the CEO was changed, but TMT remains relatively stable, avoiding the CEO appointment of TMT. This method can better describe the relationship between CEOs' functional experience and company performance.

Second, Instrumental Variable Analysis. Although the fixed effect model can solve the problem of missing variables to a certain extent. There may still be two-way causal problems. Without verification, we cannot rule out the possibility of a two-way causal relationship between firm performance and the CEO's functional experience. For example, due to the improvement of the firm's performance, it is possible to achieve a leading position in the whole industry or even more extensive cross-industry cooperation. As a result, CEOs have more opportunities for cross-border cooperation, thus enriching their functional experience. Of course, the above inference is only a possible guess. As for the possible two-way causal problem, this paper uses the instrumental variable method to test it further. An appropriate instrumental variable needs to meet two basic requirements: it is related to an endogenous variable (func_rich), and it is independent of other variables that cannot be observed but will affect the dependent variable (firm performance, ROA). This paper uses the average CEOs' functional richness (func_mean) in the same province and industry in the same year as an instrumental variable of the func_rich.

Table 5 Regression results of instrumental variable

Table 5 reports the regression results using the instrumental variable. Column (1) of Table 5 reports the regression results of the first stage of instrumental variables. It can be seen from column (1) of Table 5 that func_mean regression coefficient is significantly positive at 1%, meeting the correlation conditions. At the same time, Cragg Donald Wald F statistic result shows 1736.93, and Kleibergen Paap Wald rk F statistic result shows 1006.22, both of which reject the hypothesis of weak instrumental variables. Column (2) of Table 5 reports the regression results of the second stage of instrumental variables. It can be seen from column (2) of Table 5 that func_rich regression coefficient is significantly positive at the level of 1%, which indicates that the research conclusion of this paper is still valid after using instrumental variables to solve potential endogenous problems.

Thank you for your patiently reviewing the manuscript. We are sincerely sorry for our carelessness. We have already corrected the description according to your valuable suggestion in the revised manuscript. We hope that you will be satisfied with the revision. Subsequently, we look forward that the revised manuscript being adaptable for the demands for open publication.

I want to re-submit this manuscript and hope it is acceptable for publication in the journal. If there are any questions about our paper, please do not hesitate to let us know. Thank you very much for your attention to our paper.

Best regards,

Xiaohong Huang

School of Logistics and Management Engineering

Yunnan University of Finance and Economics

Kunming, 650221, China

E-mail: huangxiaohong@163.sufe.edu.cn

---

## [Editor Report · Decision Letter 2]

3 Feb 2023

CEO's Functional Experience and Firm Performance Based on Text Mining

PONE-D-22-03262R2

Dear Dr. Huang,

We’re pleased to inform you that your manuscript has been judged scientifically suitable for publication and will be formally accepted for publication once it meets all outstanding technical requirements.

Kind regards,

Vu Quang Trinh, PhD

Academic Editor

PLOS ONE

Additional Editor Comments (optional):

Thank you very much for your great efforts and well responses to our comments. I am happy with the work now and would like to give it a final acceptance in its current form. I hope and believe the paper will receive a good citation in the future.
---

## [Editor Report · Acceptance letter]

9 Mar 2023

PONE-D-22-03262R2 

CEO's Functional Experience and Firm Performance Based on Text Mining 

Dear Dr. Huang:

I'm pleased to inform you that your manuscript has been deemed suitable for publication in PLOS ONE. Congratulations! Your manuscript is now with our production department. 

Kind regards, 

on behalf of

Dr. Vu Quang Trinh 

Academic Editor

PLOS ONE